# High SARS-CoV-2 incidence and asymptomatic fraction during Delta and Omicron BA.1 waves in The Gambia

Sheikh Jarju[1,6], Rhys D. Wenlock [1,6], Madikoi Danso[1], Dawda Jobe[1], Ya Jankey Jagne[1], Alansana Darboe[1], Michelle Kumado[1], Yusupha Jallow[1], Mamlie Touray[1], Ebrima A. Ceesay[1], Hoja Gaye[1], Biran Gaye[1], Abdoulie Tunkara[1], Sheriff Kandeh[1], Marie Gomes[1], Ellen Lena Sylva[1], Fatoumata Toure[1], Hailey Hornsby [2,3], Benjamin B. Lindsey [2,3], Martin J. Nicklin[2,3], Jon R. Sayers[2,3], Abdul K. Sesay[1], Adam Kucharski[4], David Hodgson [4], Beate Kampmann[1,5] ✉ & Thushan I. de Silva [1,2,3] ✉

Little is known about SARS-CoV-2 infection risk in African countries with high levels of infection-driven immunity and low vaccine coverage. We conducted a prospective cohort study of 349 participants from 52 households in The Gambia between March 2021 and June 2022, with routine weekly SARS-CoV-2 RT-PCR and 6-monthly SARS-CoV-2 serology. Attack rates of 45% and 57% were seen during Delta and Omicron BA.1 waves respectively. Eighty-four percent of RT-PCR-positive infections were asymptomatic. Children under 5-years had a lower incidence of infection than 18-49-year-olds. One prior SARS-CoV-2 infection reduced infection risk during the Delta wave only, with immunity from ≥2 prior infections required to reduce the risk of infection with early Omicron lineage viruses. In an African population with high levels of infection-driven immunity and low vaccine coverage, we find high attack rates during SARS-CoV-2 waves, with a high proportion of asymptomatic infections and young children remaining relatively protected from infection.

Many African countries experienced a different COVID-19 pandemic to most high-income countries (HIC), with lower reported case numbers, hospitalisations, and deaths[1]. Underreporting due to limited testing may, in part, explain these findings, with SARS-CoV-2 seroprevalence studies suggesting high infection rates during the first year of the pandemic in African settings[2]. Non-pharmaceutical interventions (NPI) to limit SARS-CoV-2 transmission were also variably enforced and vaccine inequity has resulted in lower SARS-CoV-2 vaccination coverage in Africa compared to the highly vaccinated populations in most HIC[3].

The emergence of SARS-CoV-2 Delta and Omicron variants posed additional challenges as they displayed enhanced transmissibility and immune evasion[4–6]. Much of our understanding of these novel variants comes from studies conducted in HICs, where NPI and booster vaccines were deployed to try and limit their spread and impact on healthcare systems. There are few studies from low- and middle-income countries (LMIC) with low vaccine coverage, especially from Africa, to establish to what extent prior infection-induced immunity protected against these variants. Prospective, longitudinal, community cohort studies such as PHIRST-C in South Africa (conducted

[1]Vaccines and Immunity Theme, Medical Research Council The Gambia at the London School of Hygiene and Tropical Medicine, PO Box 273 Banjul, The Gambia. [2]Division of Clinical Medicine, School of Medicine and Population Health, The University of Sheffield, Beech Hill Road, Sheffield, UK. [3]The Florey Institute of Infection, The University of Sheffield, Sheffield, UK. [4]Centre for Mathematical Modelling of Infectious Diseases, London School of Hygiene and Tropical Medicine, Keppel Street, London, UK. [5]Institute for International Health, Charité Universitätsmedizin, Berlin, Germany. [6]These authors contributed equally: Sheikh Jarju, Rhys D. Wenlock. ✉e-mail: beate.kampmann@charite.de; t.desilva@sheffield.ac.uk

during circulation of the Beta and Delta variants[7]), which deployed regular screening irrespective of symptoms, are required to quantify true incidence and asymptomatic fraction of SARS-CoV-2 infections. Based on the modelling of serology data, attack rates of 44–81% during the first Omicron wave in South Africa were estimated, despite high seroprevalence and cumulative attack rates during prior SARS-CoV-2 waves[8]. While this suggests a greater re-infection rate with Omicron viruses, no studies have directly assessed how prior infection-induced immune history protects against Delta compared to Omicron variants in an African setting.

The first COVID-19 case in The Gambia was imported from the UK and reported on 17 March 2020[9]. Following this bans on travel and social gatherings, school closures, and compulsory wearing of face masks were implemented in The Gambia from 27 March 2020 to 17 September 2020, after which time no further NPI was enforced. Few symptomatic cases occurred until the first SARS-CoV-2 wave, which peaked in August 2020, with further waves peaking in March 2021 (driven partly by the Alpha variant), July 2021 (Delta), and January 2022 (Omicron BA.1)[10]. By July 2022, only 17% of the Gambian population had received at least one dose of SARS-CoV-2 vaccine[11].

We completed a longitudinal household cohort study in The Gambia to estimate the incidence of SARS-CoV-2 infection, the proportion of asymptomatic infections, attack rates, and factors associated with protection during the Delta (7 July 2021–4 December 2021) and Omicron BA.1 (4 December 2021 to end of study) waves, and transmission within households.

## Results

### Study population
Between 2 March 2021 and 13 June 2022, we completed a prospective, longitudinal, observational household cohort study of SARS-CoV-2 incidence and immunity in The Gambia (Transmission of Respiratory Viruses in Household in The Gambia: TransVir; clinicaltrials.gov NCT05952336). The Gambia is a small country in West Africa, ranked 174th by the United Nations Human Development Index in 2021. The climate is sub-tropical, with a short rainy season from June to October each year. The study was conducted at two urban sites, the West Coast Region and Kanifing Municipality of The Gambia. Both sites had

households that had previously participated in studies conducted at the Medical Research Council Unit The Gambia at The London School of Hygiene & Tropical Medicine (MRCG). These sites were selected due to established relationships with these communities that facilitated participant recruitment during the COVID-19 pandemic.

349 participants were recruited from 52 households, a median of 6 individuals per household (IQR 5–8); Table 1. Forty-one participants were children under 5 years old (11.7%), 153 were 5–17 years old (43.8%), 130 were 18–49 years old (37.2%) and 25 were ≥50 years old (7.2%); with 201 (57.6%) female participants. Recruitment commenced during the 2nd SARS-CoV-2 wave in The Gambia (Fig. 1), with baseline (V1) visits completed prior to the Delta (3rd) wave and most 6-month (V2) visits completed prior to the Omicron BA.1 (4th) wave[10].

Binding antibody responses to SARS-CoV-2 Spike (S) and Nucleocapsid (N) targets were measured at V1, V2 and V3 (12-month visit). SARS-CoV-2 spike antibody seropositivity increased from 56% at V1 to 84% at V2 and 94% at V3 (Table 1). A similar pattern was observed for antibodies to nucleocapsid (47% at V1, 69% at V2, 88% at V3). No participants had received SARS-CoV-2 vaccines at study entry. By 6 months of follow-up, 32 participants (6.9%) had been vaccinated, increasing to 48 participants at 12 months (14%). A median of 45 scheduled visits was completed per participant (IQR 39–48), with 338 (96%) completing more than 5 (10%) scheduled visits and included in further analyses.

### Incidence and risk factors for SARS-CoV-2 infection
The prespecified primary objectives were to establish the incidence of SARS-CoV-2 infection during the study period and to determine differences in risk between children and adults. Secondary objectives were to determine the secondary attack rate of SARS-CoV-2 in households, the sero-incidence and cumulative seroprevalence of SARS-CoV-2, the frequency of asymptomatic SARS-CoV-2 infection, and risk factors associated with SARS-CoV-2 infection.

Combined throat and nose flocked swabs (TNS) were collected for 52 weeks regardless of symptom status for SARS-CoV-2 reverse-transcriptase polymerase chain reaction (RT-PCR). Of 14569 visits with an RT-PCR result, 334 were positive (Fig. 2). A SARS-CoV-2 RT-PCR-confirmed episode was defined as ≥1 SARS-CoV-2 RT-PCR-positive TNS, with no previous RT-PCR-positive sample in the prior 28 days. Increases in antibody titre between sequential bleeds (V1–V2, V2–V3) in the absence of a positive RT-PCR result were used to define infection episodes missed by RT-PCR, with only nucleocapsid antibodies assessed for participants vaccinated between bleeds. One hundred and eleven PCR-negative but serologically-defined infection episodes were identified, equalling a total of 381 infection episodes (270 PCR-positive; 111 PCR-negative) and an incidence rate of 1.34 infections-per-person-year (95% CI 1.21–1.48).

There were 270 first infections and 111 re-infections (99 second and 12 third infections) during study follow-up. Of the 270 RT-PCR-positive infections, 236 were RT-PCR-positive during one visit only, 18 were positive for 7-14 days, 12 were positive for 14–21 days, and 4 episodes were positive for 21 days (maximum 28 days). SARS-CoV-2 infections were stratified into 'pre-Delta', 'Delta' and 'Omicron' calendar periods based on weeks when >50% of sequenced isolates in The Gambia were first Delta (7 July 2021) or Omicron-lineage (4 December 2021) viruses[10]. Attack rates were 11.8% (95% CI: 8.6–15.7) during the Pre-Delta period included in the study (prior to 7 July 2021), 44.6% (95% CI: 39.3–50.3) during the Delta period (7 July 2021 to 4 December 2021) and 56.7% (95% CI: 51.0–62.3) during the Omicron period (after 4 December 2021). In addition, Omicron had a higher attack rate amongst previously infected participants (56.3%, 95% CI: 50.3– 62.2) compared to Delta (26.7%, 95% CI: 21.6–32.3). Seven RT-PCR-positive episodes at baseline were excluded from incidence-based analyses as the time of infection was not known.

**Table 1 | Summary of demographic, serological and vaccination status of participants**

| Factor | Levels | N (%) |
|---|---|---|
| Age | <5 years | 41 (11.7%) |
| | 5–17 years | 153 (43.8%) |
| | 18–49 years | 130 (37.2%) |
| | ≥50 years | 25 (7.2%) |
| Sex | Female | 201 (57.6%) |
| | Male | 148 (42.4%) |
| Anti-spike antibody positivity[a] | V1 (n = 348) | 196 (56.3%) |
| | V2 (n = 298) | 251 (84.2%) |
| | V3 (n = 268) | 254 (94.8%) |
| Anti-nucleocapsid antibody positivity[a] | V1 (n = 348) | 165 (47.4%) |
| | V2 (n = 298) | 206 (69.1%) |
| | V3 (n = 268) | 237 (88.4%) |
| Vaccinated at 6 months (V2) | Yes | 32 (9.2%) |
| Vaccinated at 12 months (V3) | Yes | 48 (13.8%) |
| Household size, median (IQR) | – | 7 (6–10) |
| Number of rooms, median (IQR) | – | 7 (4–8) |

N = 349 unless otherwise stated.
[a]Spike and nucleocapsid antibody tested in participants who attended V1 (baseline), V2 (6-month), and V3 (12-month) visits and provided blood samples (number tested shown). No participants had been vaccinated against SARS-CoV-2 at study entry (V1).

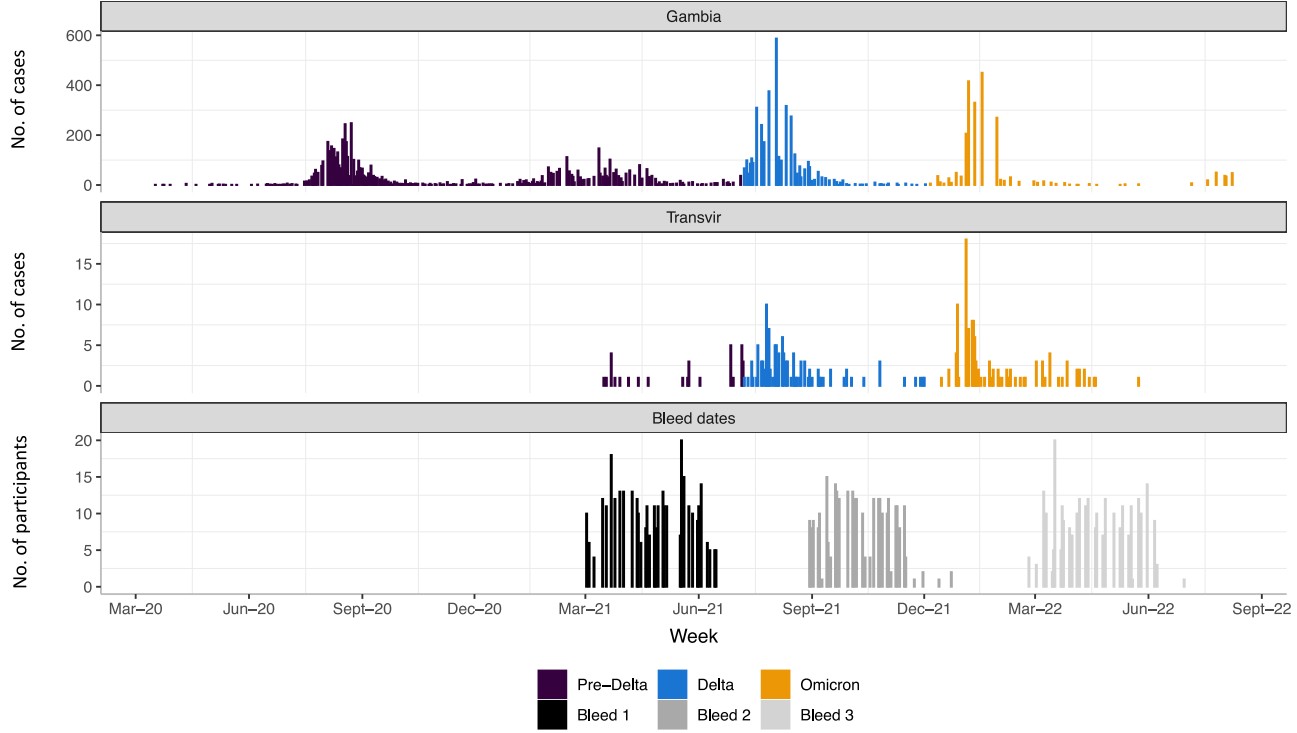

**Fig. 1 | SARS-CoV-2 epidemic curves in The Gambia in relation to study follow-up and sampling.** Top panel shows case numbers reported in The Gambia (WHO COVID-19 Dashboard. Geneva: World Health Organization, 2020. Available online: https://covid19.who.int/). Middle panel shows RT-PCR-positive events in the study cohort. Cases are coloured by SARS-CoV-2 variant period (pre-Delta, Delta, Omicron), based on dates derived from wider sequence data from The Gambia, corresponding to weeks when >50% of sequenced isolates were first Delta (7 July 2021) or Omicron-lineage (4 December 2021) viruses. Bottom panel shows a number of individuals providing samples for serological analysis during V1 (bleed 1; baseline), V2 (bleed 2; 6-month visit) and V3 (bleed 3; 12-month visit) timepoints. RT-PCR reverse-transcriptase polymerase chain reaction.

After adjusting for prior infection status, variant period, vaccination status, and household size, children under 5 years old had a lower hazard of infection than 18–49-year-olds (aHR 0.48, 95% CI: 0.31–0.74, $p = 0.0002$, Table 2). Children <5 years old had a lower hazard of infection compared to 18-49-year-olds in both Delta- and Omicron periods. These results were consistent in all sensitivity analyses using only RT-PCR-positive infection episodes, different serological definitions for RT-PCR-negative infections, and considering all RT-PCR-positive events within 90 days of a first positive result part of the same infection episode (Tables S2–S4).

V1 spike antibody status was used to assign the number of prior infections at the start of the study, with the presence of spike antibody at V1 assumed to be from one prior infection. During the study follow-up, prior infection status increased following an RT-PCR-confirmed infection or the estimated date of an RT-PCR-negative, serologically defined infection. In univariate and multivariate analyses, there was strong evidence that prior SARS-CoV-2 infection was associated with a reduced hazard of subsequent infection ($p < 0.0001$, Table 2, Fig. 3A). Compared to those previously uninfected, 1 prior infection was associated with a 58% lower hazard of infection (adjusted hazard ratio (aHR) 0.42, 95% confidence interval, CI, 0.32–0.56), with ≥2 prior infections associated with an 87% reduction in infection hazard (aHR 0.13, 95% CI: 0.09–0.20, $p < 0.0001$).

The association between prior infection status and infection hazard was modified by the dominant variant ($p$-value for interaction = 0.0002). Compared to previously uninfected individuals, those with one prior infection had a lower hazard of infection in the Delta period (aHR 0.28, 95% CI: 0.18–0.42) but not in the Omicron period (aHR 0.74, 95% CI: 0.49–1.13). In contrast, those with ≥2 prior infections had a lower hazard of infection in both Delta- and Omicron-variant periods (Table 2, Fig. 3B). These findings were consistent in all sensitivity analyses (Tables S2–S4).

To assess whether the increased infection hazard during Omicron was due to a longer interval between infections (and therefore increased waning of immunity) an additional analysis was conducted. The time elapsed between consecutive infection episodes was calculated, with a date of first infection imputed by probability sampling for those sero-positive at baseline. Infection status was categorised as: "No prior infections", "1 prior infection within 90 days", "1 prior infection 90 or more days ago", "2 or more prior infections, with the most recent within 90 days", and "2 or more prior infections, with the most recent 90 or more days ago". Compared to participants with no prior infections, participants with 1 prior infection in the last 90 days had a decreased infection hazard during the Delta (aHR 0.29, 95% CI 0.12–0.67) but not during the Omicron wave (aHR 0.73, 95% CI 0.28–1.91, Table S1). Similarly, for participants with 1 prior infection that occurred more than 90 days ago, there was a decreased infection hazard during the Delta (aHR 0.28, 95% CI 0.19–0.43) but not during the Omicron wave (aHR 0.75, 95% CI 0.50–1.14, Table S1). There remained strong evidence of interaction between the incidence of infection, prior infection status, and the dominant variant ($p < 0.0001$).

## Symptom status of RT-PCR-positive infections

At each visit, the presence of influenza-like symptoms in the preceding 7 days was assessed. A symptomatic SARS-CoV-2 episode was defined as an RT-PCR-positive episode associated with the presence of ≥1 symptom consistent with COVID-19 from 1 week before to 1 week after the first RT-PCR-positive visit. Of 263 RT-PCR-positive infections during follow-up, only 41 (15.6%) were associated with symptoms (6 missing symptom data). Cough was the most frequently reported symptom (25/41), with headache (16/41) and fever (15/41) also common (Table S5). Prior infection status, age, and variant period were not associated with the presence of symptoms (Tables S6 and S7).

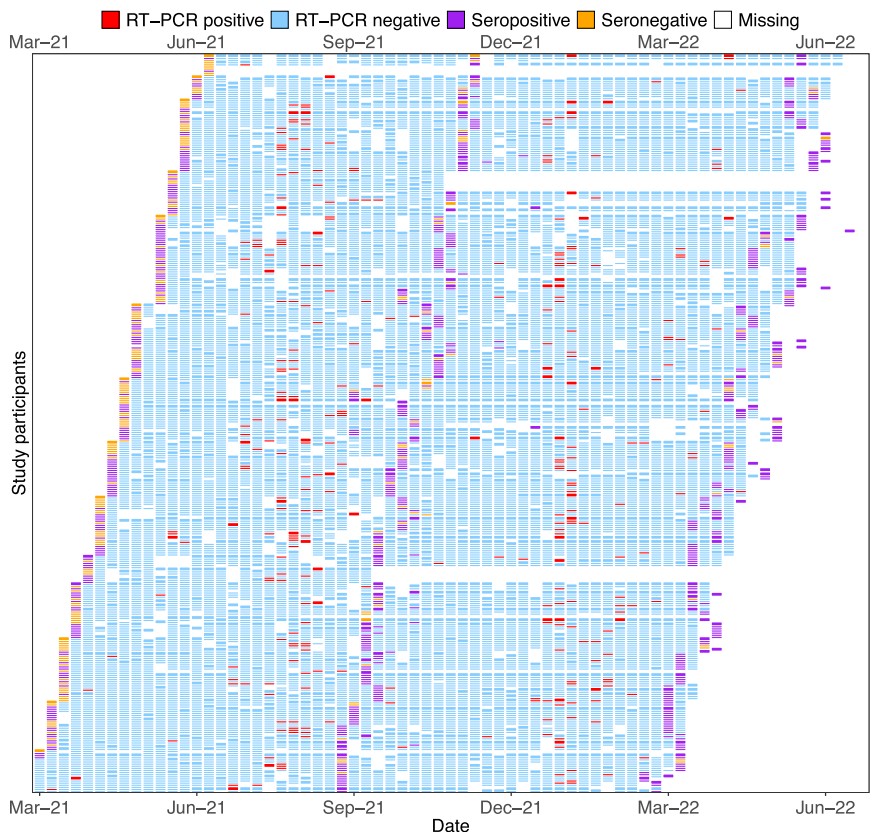

**Fig. 2 | Results of SARS-CoV-2 anti-spike serology and SARSCoV-2 RT-PCR in participants during study follow-up.** Each row represents a participant numbered consecutively, grouped together in households. Columns represent follow-up visits. A participant's follow-up time was divided into weekly breaks. Given the asynchronicity of recruitment, visits are not necessarily aligned with the same weekly breaks for all participants. As a result, a small number of weekly breaks for some participants may contain two visits. In such a situation, the result of the visit is determined hierarchically (seropositive, seronegative, RT-PCR-positive, RT-PCR-negative, missing). Seropositive and seronegative relate to positive and negative results in a serum anti-spike antibody assay. RT-PCR reverse-transcriptase polymerase chain reaction.

## Household cumulative infection risk (HCIR)

A SARS-CoV-2-positive household cluster included all individuals testing SARS-CoV-2 RT-PCR-positive in a household within ≤14 days of any RT-PCR-positive episode. Households experienced a median of 2 clusters during the study period (IQR: 1 - 3, maximum 5), with 51/52 households having at least 1 cluster. One hundred and eighteen household clusters were identified during the study, with 93 resulting from a single index case (the first RT-PCR-positive participant in a cluster) and 25 with multiple index cases (two or more participants testing RT-PCR-positive on the first day of a cluster). Of the 93 single index cases initiating clusters, 14 (16%) were symptomatic (5 missing symptom data), 3 were under 5 years old, 46 were 5–17 years old, 38 were 18–49 years old and 6 were >50 years old. 466 participants were exposed in the 93 single-index clusters, with 43 subsequently testing SARS-CoV-2 RT-PCR-positive (an HCIR of 9.2%). Four hundred and forty-four participants underwent RT-PCR testing within one serial interval (14 days) of the index case positive RT-PCR, of which 35 tested positive (a secondary attack rate, SAR, of 7.9%). There was no evidence of any association with characteristics of the index case, variant period, or household with an increased odds of transmission to exposed participants (Tables S8 and S9). There was weak evidence in the univariate HCIR analysis that positive serology in the index case was associated with lower odds of onward transmission compared to indexes with negative serology (OR 0.47, 95% CI 0.21– 0.99, $p = 0.05$, Table S8). This finding was not replicated in the univariate SAR analysis (OR 0.57, 95% CI 0.24–1.25, $p = 0.16$, Table S10). After adjusting for the SARS-CoV-2 serology status of the exposed household contacts, no association remained in either HCIR (aOR, 0.57, 95% CI 0.22–1.14, $p = 0.06$) or SAR analyses (aOR 0.57, 95% CI 0.23–1.30, p = 0.23, Tables S9 and S11).

## Discussion

We report the incidence of symptomatic and asymptomatic SARS-CoV-2 from a prospective, 52-week household cohort study conducted in The Gambia during the emergence and circulation of the Delta and Omicron SARS-CoV-2 variants. In a population with low vaccine coverage (14%) by the end of the study, we report high population attack rates during Delta (45%) and Omicron (57%) waves despite high seroprevalence prior to each wave (56% spike seropositivity pre-Delta and 84% pre-Omicron). We find that one prior infection was protective against infection during the Delta wave only, with immunity from ≥2 prior infections required to provide protection from infection during the Omicron wave. Interestingly, children aged <5 years remained relatively protected from infection in both Delta and Omicron waves, and only 16% of SARS-CoV-2 infections were symptomatic.

SARS-CoV-2 infection and mortality rates in many African countries were lower than expected[12]. Mounting evidence suggests that disease severity and mortality may have been genuinely lower in some African settings compared to most HICs. No overall significant excess mortality was observed in The Gambia during 2020, prior to the Delta wave[13]. As sampling in our study was not symptom-driven, we were able to characterise the percentage of asymptomatic cases in RT-PCR-confirmed infections during follow-up, which was high at 84%. A meta-analysis of global studies during the first COVID-19 pandemic year concluded that the overall proportion of asymptomatic SARS-CoV-2 infections was 24%[14], with a recent systematic review reporting that the

**Table 2 | The association between prior infection status, SARS-CoV-2 variant period, age and household size on the incidence of SARS-CoV-2 infection**

| | Number of participants (%) | Number of infections | Incidence rate (95% CI) | Crude hazard ratio (95% CI) | p-value | Adjusted hazard ratio (95% CI)a | p-value | Delta-specific aHR (95% CI)b | Omicron-specific aHR (95% CI)b | p-value for interaction |
|---|---|---|---|---|---|---|---|---|---|---|
| **Prior infection** | | | | | | | | | | |
| 0 | 144 (42.6) | 118 | 1.96 (1.64–2.35) | ref | | ref | | ref | ref | |
| 1 | 310 (91.7) | 205 | 1.37 (1.19–1.57) | 0.70 (0.58–0.84) | <0.0001 | 0.42 (0.32–0.56) | <0.0001 | 0.28 (0.18–0.42) | 0.74 (0.49–1.13) | |
| ≥2 | 204 (60.4) | 51 | 0.73 (0.56–0.97) | 0.38 (0.29–0.51) | <0.0001 | 0.13 (0.09–0.20) | <0.0001 | 0.05 (0.02–0.14) | 0.23 (0.14–0.36) | 0.0002 |
| **Period** | | | | | | | | | | |
| Pre-Delta | 338 (100) | 30 | 0.45 (0.32–0.65) | ref | | | | | | |
| Delta | 334 (98.8) | 155 | 1.33 (1.14–1.56) | 3.02 (2.03–4.49) | | | | | | |
| Omicron | 292 (86.4) | 189 | 1.95 (1.69–2.25) | 4.93 (3.43–7.09) | <0.0001 | | | | | |
| **Age (years)** | | | | | | | | | | |
| <5 | 39 (11.5) | 34 | 1.07 (0.76–1.5) | 0.80 (0.60–1.07) | | 0.48 (0.31–0.74) | | 0.54 (0.30–0.98) | 0.43 (0.25–0.74) | |
| 5–17 | 151 (44.7) | 173 | 1.37 (1.18–1.59) | 1.00 (0.86–1.17) | | 0.84 (0.64–1.10) | | 0.93 (0.64–1.36) | 0.75 (0.54–1.03) | |
| 18–49 | 123 (36.4) | 137 | 1.36 (1.15–1.61) | ref | | ref | | ref | ref | |
| ≥50 | 25 (7.4) | 30 | 1.43 (1–2.04) | 1.08 (0.83–1.40) | 0.61 | 1.07 (0.72–1.59) | 0.0002 | 1.31 (0.71–2.41) | 0.86 (0.51–1.46) | 0.64 |
| **Household size** | | | | | | | | | | |
| 5–7 | 124 (36.7) | 144 | 1.39 (1.18–1.64) | ref | | ref | | | | |
| 8–10 | 125 (40.0) | 136 | 1.28 (1.08–1.52) | 0.93 (0.79–1.09) | | 1.06 (0.83–1.36) | | | | |
| >10 | 89 (26.3) | 94 | 1.35 (1.10–1.65) | 0.98 (0.82–1.17) | 0.81 | 1.05 (0.78–1.41) | 0.87 | | | |

P-values were calculated through likelihood ratio tests of nested models and are two-sided. All models use the Anderson–Gill extension of the proportional Cox hazards model, accounting for clustering by participant and household. The number of infections include both RT-PCR-positive and serologically defined RT-PCR-negative infections (defined by either (i) seroconversion of spike antibody from negative to positive or (ii) boosting of spike and nucleocapsid antibody above the median value seen in RT-PCR-positive episodes in unvaccinated individuals and nucleocapsid antibody alone in vaccinated individuals.
CI confidence interval, ref reference level, aHR adjusted hazard ratio.
aAdjusted for prior infection status, age, household size and vaccination status stratified by period.
bCalculated through the incorporation with an interaction term between the variable of interest and period with adjustment for prior infection status, age, household size and vaccination status.

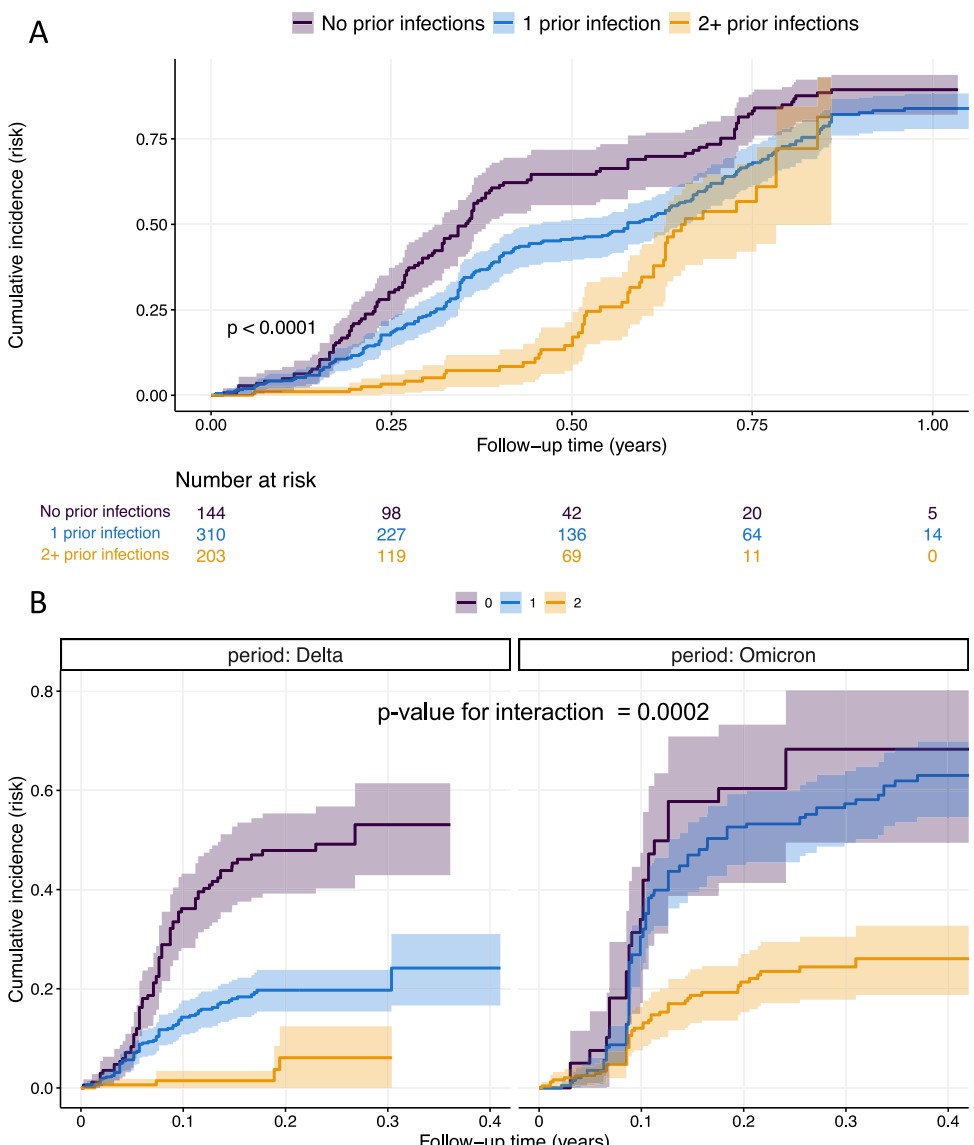

**Fig. 3 | The cumulative incidence of SARS-CoV-2 infection by prior infection status.** The cumulative incidence over follow-up time is expressed as a risk (number infected/number at-risk) and displayed for **A** the whole study period or **B** stratified by Delta- and Omicron-variant periods. *P*-value calculated by log-rank test for association between prior infection status and infection incidence (**A**) and for interaction between Delta and Omicron periods (**B**). The test for interaction assesses whether the association between prior infection status and infection incidence is modified by the circulating variant (i.e., is the effect of one prior infection on infection incidence different between Delta and Omicron). All *p*-values are two-sided. Shaded area represents the 95% confidence interval.

pooled proportion of asymptomatic cases in Omicron infections was similar at 26%[15]. A prospective household cohort study in South Africa (PHIRST-C) with a similar design to ours showed a remarkably similar percentage of asymptomatic infections (85%), with equivalent estimates across D614G (87%), Beta (84%) and Delta (82%) infections[7]. Pooled estimates from 12 studies from Africa before the end of 2020 showed a higher asymptomatic percentage (64%) than those from America (40%), Asia (18%) and Europe (28%)[14]. The main reason for this milder clinical presentation of SARS-CoV-2 in Africa may be the younger age profile in most African countries. The median age in The Gambia is 17 years, compared to 40 years in the UK and 38 years in the USA[16,17]. While we did not find an association between age and symptomatic illness in our study, PHIRST-C reported that the percentage of symptomatic SARS-CoV-2 was 9.2% in children <5 years compared to 35% in adults ≥60 years[7]. Additional factors may also contribute to higher asymptomatic SARS-CoV-2 infections in Africa, such as greater cross-reactive immunity from prior coronavirus infections[9,18,19] and

higher frequencies of T-cell responses associated with mild or asymptomatic infection[20].

Prior to the circulation of Omicron BA.1 in The Gambia, substantial population immunity had accumulated through infections during three waves[10]. Despite this, we estimated an attack rate of at least 57% during the 4th wave caused primarily by Omicron BA.1, with individuals requiring immunity from ≥2 prior infections to provide protection against infection during this first Omicron surge. The substantial antibody immune escape demonstrated by Omicron lineage viruses likely underpins this higher likelihood of breakthrough infections[21]. A recent meta-analysis estimated that while past infection was highly effective against re-infection with ancestral, Alpha, Beta and Delta variants, the pooled effectiveness against symptomatic disease with Omicron viruses was much lower at 45%[22]. Our Omicron BA.1 attack rate estimated by a combination of RT-PCR and serology is very similar to those based on serological modelling from urban (58%) and rural (65%) South Africa in the PHIRST-C cohorts[23]. Further exploration of

immune correlates of protection in our cohort will be valuable, with a particular focus on mucosal immunity, which is induced by infection but less consistently by parenteral vaccines[24,25].

A striking finding in our cohort was that children <5 years old were relatively protected from SARS-CoV-2 infection, even with the more transmissible Omicron BA.1 variant. Many studies have reported that young children are protected from severe COVID-19 and are more likely to experience mild or asymptomatic infection[26–29]. Our data suggest that young children may be relatively protected from SARS-CoV-2 infection and not just severe disease. Future studies should establish the mechanisms that result in this protective phenotype. Potential hypotheses include stronger innate antiviral responses in the upper airways of children potentially primed by other respiratory viruses, local microbiome differences, and epigenetic alternations in innate immune cells[26,30,31].

Our study has several limitations worth considering. Defining re-infections in seropositive individuals based on serological boosting in the absence of a positive RT-PCR is challenging due to variable waning and boosting parameters and long intervals between serology sampling timepoints. Similar assumptions and limitations are relevant when imputing infection dates in individuals found to be sero-positive at baseline, used to incorporate time from the last infection in our models. Nevertheless, our main findings were consistent across multiple sensitivity analyses incorporating different thresholds for defining these RT-PCR-negative infections. However, it is possible that infections in young children were more likely to be missed by RT-PCR *and* result in lower or absent serological response, making it more likely that their infections were missed.

Our HCIR (9.2%) and SAR (7.9%) estimates could also only be based on RT-PCR-positive events and are likely to be significant under-estimates. Despite this, we found weak evidence of an association between the serology of the index case and onward transmission to exposed household contacts, although this finding was not significant when accounting for serostatus in contacts and is likely to be limited by the small number of transmissions detected. Given the low rate of vaccination in our population, we were unable to evaluate the impact of vaccination on SARS-CoV-2 incidence risk and were also unable to collect information on the specific vaccines and the number of doses received. However, based on discussions with vaccination services in The Gambia, we believe that most vaccinees received a single dose of J&J/Janssen's Ad26.COV2.S vaccine. Finally, our definitions of Delta and Omicron infections were based on wider sequencing data from The Gambia rather than sequencing of individual infections in our study, so some misclassification could have occurred during pre-Delta to Delta and Delta to Omicron cross-over periods.

With no ongoing SARS-CoV-2 vaccination programmes in many African countries, questions remain around the breadth and potency of immunity that continues to be shaped by repeated SARS-CoV-2 infections and what implications there are for epidemic surges with new variants each year. While the high proportion of asymptomatic infections seen in The Gambia is reassuring, the emergence of SARS-CoV-2 variants with greater severity may be a cause for concern in the face of high population attack rates due to sub-optimal immunity. Furthermore, how immunity to SARS-CoV-2 will build during child-hood through repeated infections is important to characterise and to establish whether this will result in mild SARS-CoV-2 in adulthood (similar to seasonal coronavirus infections), even in the absence of vaccination.

## Methods

### Study participants and recruitment
Households expressing interest in joining the study were invited to attend MRCG for informed consent and screening. All households with ≥5 consenting members, including at least one adult and one child, were eligible. Written (or thumb-printed) informed consent was obtained from all adult participants, assent was obtained from children aged 12–18 years, and parents or guardians provided consent for children <12 years old. The study was approved by the joint Gambia Government and MRCG ethics committee, and the London School of Hygiene and Tropical Medicine ethics committee (ID 22556).

Households were recruited in a staggered manner (approximately 4/week). Participants underwent three scheduled clinic visits at base-line (V1), 6 months (V2), and 12 months (V3), and weekly home collection of combined throat and nose flocked swabs (TNS) for 52 weeks. All TNS were transported on ice packs to MRCG laboratories on the same day and stored at −70 °C until further use. Participants were provided with a mobile phone and encouraged to contact the study team if they experienced a fever, cough, or shortness of breath (defined as influenza-like illness, ILI) and were promptly seen for an unscheduled visit, where a TNS was collected and tested within 24 h for SARS-CoV-2 by reverse-transcriptase polymerase chain reaction (RT-PCR). At scheduled clinic visits (V1-V3), participants provided a TNS, whole blood collected for peripheral blood mononuclear cell and serum separation, and nasal lining fluid collected using a synthetic absorptive matrix strip. Household- and individual-level socio-demo-graphic data were collected at each clinic visit, including SARS-CoV-2 vaccination status. HIV prevalence is low (1.4%) in The Gambia[32] and in accordance with standard practice in research studies, we did not test for HIV infection as part of our study. At each weekly visit, the presence of symptoms consistent with ILI/COVID-19 during the previous 7 days was recorded. All data were entered into a REDCap database. The full study protocol is available as supplementary information.

### SARS-CoV-2 RT-PCR
Ribonucleic acid (RNA) was extracted from each TNS using the QIAamp Viral RNA kit (QIAGEN, Germany, cat. no. 52904 and 52906) as per the manufacturer's instruction, followed by SARS-CoV-2 RT-PCR as previously described[33,34]. A 4-plex assay detecting SARS-CoV-2 envel-ope (E) and nucleocapsid (N) genes, along with RNAseP and an internal control (Phocine herpes virus; PhHV) was used. A positive SARS-CoV-2 RT-PCR was defined as samples where all four targets had a cycle threshold value of ≤37. Samples were repeated when only one of the SARS-CoV-2 targets was detected.

### SARS-CoV-2 serology
SARS-CoV-2 S- and N-specific immunoglobulin G (IgG) was measured using a previously described in-house enzyme-linked immunosorbent assay (ELISA), shown to have 99.5% sensitivity and 98.8% specificity for anti-S IgG, and 99.5% sensitivity and 84.1% specificity for anti-N IgG[35]. Immunolon 4 HBX microplates (Thermo Scientific, USA, cat no. 7350465) were coated with either the full-length extracellular domain of the SARS-CoV-2 S glycoprotein produced in mammalian cells or a full-length N protein produced in *Escherichia coli*[36]. A goat anti-human IgG antibody conjugated to horse radish peroxidase (Invitrogen, 62-8420) was used at 1:500 dilution in both anti-S and -N IgG ELISA assays as the secondary antibody. A standard curve calibrated to the WHO International Standard for anti-SARS-CoV-2 Immunoglobulin (cat no. NIBSC 20/136) was used to quantify S and N antibody titres. Serostatus for each antigen was determined using previously defined thresholds based on the ratio between the sample and control well optical densities[35].

### Definitions and statistical analyses
We aimed to recruit 50–70 households, assuming a median household size of seven. This allowed for the incidence to be estimated with a 12% absolute precision (±6%) and would provide an 80% power (at alpha = 0.05) to detect a 20% difference in absolute incidence risk between adults and children. A SARS-CoV-2 RT-PCR-confirmed epi-sode was defined as ≥1 SARS-CoV-2 RT-PCR-positive TNS, with no previous RT-PCR-positive sample in the prior 28 days. A sensitivity

analysis for main study outcomes was conducted with this threshold set at 90 days. A symptomatic SARS-CoV-2 episode was defined as an RT-PCR-positive episode associated with the presence of ≥1 symptom consistent with COVID-19 from 1 week before to 1 week after the first RT-PCR-positive visit. Individuals found to be SARS-CoV-2 RT-PCR-positive at the enrolment visit (V1) were excluded from the assessment of symptomatic fraction due to uncertainty around when this infection started.

Increases in antibody levels between visits (V1–V2, V2–V3) were used to define SARS-CoV-2 infections in the absence of a positive RT-PCR using several criteria: (i) for seronegative individuals, spike antibody seroconversion (negative-to-positive) between visits in the absence of a history of vaccination or (ii) for seropositive individuals, a boost in spike *and* nucleocapsid antibody titres above the median fold-change seen in RT-PCR-positive cases during the same period (V1–V2 or V2–V3) in the absence of vaccination, or a boost in nucleocapsid antibody alone if the individual was vaccinated between the time-points. Sensitivity analyses for the main study outcomes were also undertaken using different thresholds of antibody boosting to define PCR-negative infections in spike-seropositive individuals (Table S12).

The Anderson-Gill extension of the Cox proportional hazards model was used for the assessment of SARS-CoV-2 incidence. The Anderson-Gill extension allowed for participants to have recurrent events with robust estimation accounting for within-person correlation as well as intra-household correlation, given the household-based sampling employed here. This allowed for a varying baseline hazard over time, as observed with waves of SARS-CoV-2[10]. Age (<5, 5–18, 18–49 and ≥50 years), prior infection status (time-varying co-variate; 0, 1, ≥2 infections) and household size (5–7, 8–9, ≥10) were the exposures of interest. The presence of anti-spike IgG at V1 was assumed to signify one prior infection by study entry. A participant's prior infection status was enumerated as a time-varying covariate described as no prior infections, 1 prior infection and ≥2 prior infections. An additional analysis was conducted exploring the dual effect of prior infection status and the time from previous infection on infection incidence. SARS-CoV-2 vaccination status was included as a fixed effect (time-varying where the exact vaccination date is known). Due to violating the proportional hazards assumption, only crude (unadjusted) incidence rates and 95% confidence intervals (CI) are presented for the association between calendar period and infection. Calendar period was included in regression through stratification in the main adjusted models (mitigating the non-proportionality) and as a fixed effect with an interaction term with prior infection status and age for variant-specific models.

Crude and multivariate-adjusted hazard ratios and 95% CI were calculated for exposure variables. P-values were calculated through likelihood ratio tests (LRTs) comparing two nested models, one with the variable of interest. Calendar period-specific hazard ratios were calculated for the association between prior infection status/age and SARS-CoV-2 incidence. Mixed effects logistic regression, with household and participant as random effects, was used to assess risk factors for transmission after exposure to an index case (secondary attack rate). Fixed effect modelling was used to assess associations with symptomatic infections due to the small number of outcomes. Associations are presented as odds ratios with 95% CI and p-values from LRTs.

A SARS-CoV-2-positive household cluster included all individuals testing SARS-CoV-2 RT-PCR-positive in a household within ≤14 days of any RT-PCR-positive episode. The index case was defined as the first participant testing RT-PCR-positive in each cluster. The household cumulative infection risk was defined as the percentage of household members with RT-PCR-confirmed episodes in a cluster (denominator as total individuals in the household tested during the cluster). In clusters with a single index case, the secondary attack rate was calculated as the percentage of household members with SARS-CoV-2 RT-PCR-confirmed episodes following the first positive case. All significant tests are two-sided.

Analysis was conducted in R studio using the packages readxl (v1.4.3), tidyverse (v2.0), ggplot2 (v3.4.2), survival (v3.5-5), lubridate (v1.9.3), survminer (v.0.4.9), broom (v1.0.5), popEpi (v.0.4.11), and EpiR (v.2.0.62).

### Inclusion and ethics statement
This study was conducted with the inclusion of researchers from The Gambia throughout the research process. The aims and objectives of the research were considered locally relevant during study design and approved by the joint Gambia Government and MRCG ethics committee. The study team consisted of Gambian researchers working in collaboration with international partners. Scientific capacity building was central to laboratory aspects of the study, with the transfer of molecular and serological assays to The Gambia so all data could be generated in-country following the training of Gambian research scientists. All members of the research team in The Gambia who made relevant contributions are included as co-authors, including joint-first author S.J., with others listed in the Acknowledgements section.

### Reporting summary
Further information on research design is available in the Nature Portfolio Reporting Summary linked to this article.

### Data availability
Aggregate data to reproduce figures are available online at https://github.com/Rhys-wenlock/Transvir_SCV2_incidence. Individual-level data cannot by publicly shared due to ethical restrictions and the potential for identifying included individuals. To request individual participant data access, please contact the corresponding author T.d.S (t.desilva@sheffield.ac.uk) who will respond within 1 month of request. Upon approval, data can be made available through a data-sharing agreement.

### Code availability
The entire code used for analyses described in this manuscript is available online at https://github.com/Rhys-wenlock/Transvir_SCV2_incidence. The version of the code used for analysis in this manuscript can be permanently found with the DOI: 10.5281/zenodo.10955388.

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

## Acknowledgements

The study was funded by a United Kingdom Research and Innovation Grant (No. MC_PC_19084). We are extremely grateful to all the TransVir participants who gave up their time and engaged so fully with the study. We also acknowledge the wider field TransVir field team members Malang Mbenga, Fansu Dibba, Yusupha Fadera, Yusupha Faal, Dawda Jawara, Omar Jallow, Momodou Lamin Sanyang, Sarjo Wassa Koita, Hulaimatu Bangura, Many Gibba, Maryama Jawara and Fatou Sanyang for their dedication and hard work during the field study.

## Author contributions

S.J., R.D.W., M.D., A.K.S., A.K., B.K., and T.I.d.S. conceived and designed the study. S.J., R.D.W., M.D., D.J., Y.J.J., A.D., Y.J., M.T., E.A.C., H.G., B.G., A.T., S.K., M.G., E.L.S., F.T., A.K.S., B.K. and T.I.d.S. collected and processed laboratory data. S.J., R.D.W., M.D., D.J., Y.J.J., A.D., M.K., H.H., B.B.L., M.J.N., J.R.S., A.K.S., A.K., D.H., B.K. and T.I.d.S. analysed and interpreted data. S.J., R.D.W., M.D., M.K., B.K. and T.I.d.S. assessed and verified underlying data. S.J., R.D.W. and T.I.d.S. drafted the article. All authors read and approved the final draft.

## Competing interests

The authors declare no competing interests.
