## [Peer Review File · Nature Communications]

High SARS-CoV-2 incidence and asymptomatic fraction during Delta and Omicron BA.1 waves in The GambiaREVIEWER COMMENTS

Reviewer #1 (Remarks to the Author):

This is a highly informative paper that provides estimates on the burden of SARS-CoV-2 within an understudied country, Gambia. The article provides useful information regarding the incidence of SARS-CoV-2 during Delta and BA.1 periods, the symptom fraction, and the household attack rates. I believe the findings of this study are incredibly important as they highlight the continued need for SARS-CoV-2 surveillance, even in populations with high seropositivity.

Specific notes:

Overall note: I had to read through the results multiple times to figure out where everything was. I strongly recommend re-organizing the introduction and findings. Please see comments below for more details.

Introduction:

1. I think there is a bit of information missing between the first and second sentences of the second paragraph. The first sentence talks about NPIs and boosters in HCI and then the second states that there is limited information on the levels of protection provided by prior infections within LIC. I am not sure how these two sentences are linked.
2. While I understand that there is limited data on SARS-CoV-2 burdens in LIC, the statements about where the gaps seem a bit limited relative to the stated goal of the paper. The goal being to "to estimate the incidence of SARS-CoV-2 infection, the proportion of asymptomatic infections, attack rates and factors associated with protection during the Delta and Omicron BA.1 waves, and transmissions within households."
3. Please add the date ranges for the Delta and Omicron BA.1 periods to the introduction.

Results:

4. Line 93: Please spell out numbers if the sentence starts with them. (number: 41) and again on line 130 (number: 111)
5. The authors state that an increase in antibody titre between sequential bleeds without a documented PCR was defined as a missed PCR. How did the authors account for vaccinations?
6. It seems like the authors had data, though not much during the "Pre-Delta" period. I recognize that this is a small period of time with likely low transmission but I don't understand why the authors did not provide attack rates presented.
7. In the paragraph between lines 154-162, it seems like the ordering should be flipped, with the variant-specific stuff coming second and the overall infection numbers coming first.
8. I think "After adjusting for prior infection status, variant period, vaccination status, and household size, children under 5 years old had a lower hazard of infection than 18-49-year-olds (aHR 0.48, 95% CI: 0.31-0.74, $p=0.002$, Table 2)." Is oddly placed between two sections focused on prior infection as the exposure of interest.
9. The authors state that prior infection provided protection during the Delta period but not the Omicron period. This makes me wonder if there is a difference in time since the last prior infection.
10. In S7 and S8 the authors look at the association between potential index case factors and the risk of transmission. Would it be possible to include prior infection or vaccination status in these tables, thus getting at a reduction in transmissibility following immunity-conferring events?

Discussion:

11. Remove "is" from line 239 "Africa is may be"

Reviewer #1 (Remarks on code availability):

The code is well laid out in GitHub with a clear ReadMe. The code is to be run in numeric order, which is great and all figure code is present.

Reviewer #2 (Remarks to the Author):

Jarju and Wenlock et al conduct a prospective SARS-CoV-2 cohort study of 52 households in the Gambia over 52 weeks between March 2021 and June 2022, with weekly SARS-CoV-2 RT-PCR and 6-monthly SARS-CoV-2 serology. They have estimated high attack rates during the Delta and Omicron BA.1 waves, and symptomatic proportions in an African population with a low (about 14%) vaccine coverage. They have found prior infections are associated with reductions in infection risks during the delta predominance, and 2+ prior infections are required to develop protection against early omicron lineages. While this longitudinal study provides useful information on SARS-CoV-2 infection, symptomatic infection and protection provided by prior infections, certain questions remain to be addressed. Additionally, conducting further investigations are recommended to enhance the manuscript (details below).

1. "We aimed to recruit 50–70 households, assuming a median household size of seven." - Justifications on the aimed sample size are required.
2. "A cox proportional hazards model was fitted for the assessment of SARS-CoV-2 incidence with the Anderson-Gill extension accounting for clustering by individual (repeated measures) and household. This allowed for a varying baseline hazard over time, as observed with waves of SARS-CoV-2" – This is not clear. What does "clustering by individual (repeated measures)" refer to? Clarify for what purpose the Anderson-Gill extension is used.
3. "Calendar period was included in regression through stratification in all adjusted models (mitigating the non-proportionality) and as an interaction term with prior infection status and age" – The stratification and the interaction terms should not be in one model (the same model) because they could result in a similar effect (allow for varying risks of the exposure between periods). If I understand this sentence correctly, revisions on analyses are needed.
4. While I find the results of protection provided by prior infections and the variations by variants interesting, the analyses could be improved by conducting a deeper investigation, e.g., how the protection varies in relation to the time elapsed from the prior infection to the present infection.
5. Table 2 footnote, "***Calculated through the incorporation with an interaction term between the variable of interest and period with adjustment for prior infection status, age, household size and vaccination status, stratified by period." – same with comment 3.
6. "444 participants underwent RT-PCR testing within one serial interval of the index case positive RT-PCR, of which 35 tested positive (a Secondary Attack Rate of 7.9%)" – What is the serial interval used in the study? Please state. Also, does the SAR vary by the variants or period?
7. Discussion: "Despite this, we estimated an attack rate of at least 57% during the 4th wave caused primarily by Omicron BA.1, with individuals requiring immunity from ≥ 2 prior infections to provide protection against infection during this first Omicron surge. The substantial antibody immune escape demonstrated by Omicron lineage viruses likely underpins this higher likelihood of breakthrough infections" – To support the above discussion, addition of analyses to investigate the reinfection risk by variants (delta vs omicron) is suggested.
8. Discussion: "Many studies have reported that young children are protected from severe COVID-19 and more likely to experience mild or asymptomatic infection (ref 25)" – the authors mention "many studies", yet cite one article. While I understand that this article may be a review, I suggest adding references of the studies that provide data in support of this discussion. If such data are not found, suggest revising the sentence.
9. "Attack rates were 45% during the Delta period (7th July 2021 to 4th December 2021) and 57% during the Omicron period (after 4th December 2021)." – provide the confidence intervals.
10. "There were 270 first infections and 111 re-infections (99 second and 12 third infections) during study follow-up." – same with comment 7, suggest adding analyses to investigate reinfection risk by variants (delta vs omicron).

Responses to reviewers

We thank both reviewers for their constructive and helpful comments, which have served to improve our manuscript. We have detailed how each comment was addressed below.

Reviewer #1 (Remarks to the Author):

This is a highly informative paper that provides estimates on the burden of SARS-CoV-2 within an understudied country, Gambia. The article provides useful information regarding the incidence of SARS-CoV-2 during Delta and BA.1 periods, the symptom fraction, and the household attack rates. I believe the findings of this study are incredibly important as they highlight the continued need for SARS-CoV-2 surveillance, even in populations with high seropositivity.

Specific notes:

Overall note: I had to read through the results multiple times to figure out where everything was. I strongly recommend re-organizing the introduction and findings. Please see comments below for more details.

Introduction:

1. I think there is a bit of information missing between the first and second sentences of the second paragraph. The first sentence talks about NPIs and boosters in HCI and then the second states that there is limited information on the levels of protection provided by prior infections within LIC. I am not sure how these two sentences are linked.

Thank you – we have tried to improve the flow of this paragraph and have reworded as follows “The emergence of SARS-CoV-2 Delta and Omicron variants posed additional challenges as they displayed enhanced transmissibility and immune evasion. Much of our understanding of these novel variants comes from studies conducted in HICs, where NPI and booster vaccines were deployed to try and limit their spread and the impact on healthcare systems.”

2. While I understand that there is limited data on SARS-CoV-2 burdens in LIC, the statements about where the gaps seem a bit limited relative to the stated goal of the paper. The goal being to “to estimate the incidence of SARS-CoV-2 infection, the proportion of asymptomatic infections, attack rates and factors associated with protection during the Delta and Omicron BA.1 waves, and transmissions within households.”

We have expanded these statements as follows “There are few studies from low- and middle-income countries (LMIC) with low vaccine coverage, especially from Africa, to establish to what extent prior infection-induced immunity protected against these variants. Prospective, longitudinal, community cohort studies such as PHIRST-C in South Africa (conducted during circulation of the Beta and Delta variants), which deployed regular screening irrespective of symptoms, are required to quantify true incidence and asymptomatic fraction of SARS-CoV-2 infections. Based on modelling of serology data, attack rates of 44 – 81% during the first Omicron wave in South Africa were estimated, despite high seroprevalence and cumulative attack rates during prior SARS-CoV-2 waves. While this suggests a greater re-infection rate with Omicron viruses, no studies have directly assessed how prior infection-induced immune history protects against Delta compared to Omicron variants in an African setting.”

3. Please add the date ranges for the Delta and Omicron BA.1 periods to the introduction.

We have included the dates as requested (“...the Delta (7th July 2021 to 4th December 2021) and Omicron BA.1 (4th December 2021 to end of study) waves, and transmission within households.”

Results:

4. Line 93: Please spell out numbers if the sentence starts with them. (number: 41) and again on line 130 (number: 111)

Thank you for pointing out this oversight - we have corrected this here and throughout the manuscript.

5. The authors state that an increase in antibody titre between sequential bleeds without a documented PCR was defined as a missed PCR. How did the authors account for vaccinations?

For participants vaccinated between sequential bleeds their nucleocapsid antibodies (and not their spike) were used to determine RT-PCR-negative infection episodes as described in the methods. We have now also added to the sentence in the results section to make this clearer - “Increases in antibody titre between sequential bleeds (V1 to V2, V2 to V3) in the absence of a positive RT-PCR result was used to define infection episodes missed by RT-PCR, with only nucleocapsid antibodies assessed for participants vaccinated between bleeds.”

6. It seems like the authors had data, though not much during the “Pre-Delta” period. I recognize that this is a small period of time with likely low transmission but I don’t understand why the authors did not provide attack rates presented.

Thank you for highlighting this oversight – we have now included the attack rate for Pre-Delta “Attack rates were 11.8% (95% CI 8.6 – 15.7) during the Pre-Delta period included in the study (prior to 7th July 2021), 44.6% (95% CI 39.3 – 50.3) during the Delta period (7th July 2021 to 4th December 2021) and 56.7% (95% CI 51.0 – 62.3) during the Omicron period (after 4th December 2021).”

7. In the paragraph between lines 154-162, it seems like the ordering should be flipped, with the variant-specific stuff coming second and the overall infection numbers coming first.

Thank you, we agree and have re-ordered this paragraph as suggested.

8. I think “After adjusting for prior infection status, variant period, vaccination status, and household size, children under 5 years old had a lower hazard of infection than 18-49-year-olds (aHR 0.48, 95% CI: 0.31-0.74, p=0.002, Table 2).” Is oddly placed between two sections focused on prior infection as the exposure of interest.

Thank you – we have moved this paragraph to above the section on prior infection status.

9. The authors state that prior infection provided protection during the Delta period but not the Omicron period. This makes me wonder if there is a difference in time since the last prior infection.

Thank you for raising this. We have conducted an additional analysis to explore whether the increased hazard of infection seen with Omicron in participants with 1 prior infection may have been due to waning of immunity (due to their first infection being earlier/longer ago than those infected by delta). To do this we have used probability sampling (based upon Gambia-wide case numbers) to estimate the date of infection for participants who were sero-positive at our V1 baseline visit, so that we can infer a date of previous infection and subsequently a time from infection to infection. This methodology is the same as was used to impute dates of PCR-negative serologically confirmed infections.

We have then re-fitted our Proportional Hazards model with a time-varying combined prior infection variable (categories of: No prior infections, 1 prior infection within 90 days, 1 prior infection 90 or more days ago, 2+ prior infections within most recent within 90 days, 2+ prior infections with most recent 90 or more days ago). The results of this can be found in supplementary table 1.

Importantly, this additional analysis demonstrates that after adjusting for time since last infection, 1 prior infection was not sufficient for protection against Omicron infection whereas it was for Delta. Within 90 days of a participant's first infection, the HR for infection by Delta was 0.29 (95% CI 0.12 – 0.67) whereas for Omicron it was 0.73 (95% CI 0.28 – 1.91). A similar association was observed for participants with 1 prior infection 90 or more days ago, with a HR for Delta infection of 0.28 (95% CI 0.19 – 0.43) and Omicron infection of 0.75 (95% CI: 0.50 – 1.14). The confidence intervals for Omicron infection cross 1 for both infection intervals.

Please note that there are important assumptions and limitations in this analysis. This is effectively an analysis of a sub-group of a sub-group, with which the power to detect meaningful differences decreases. In addition, it is possible that we have misclassified the date of previous infection for those sero-positive at baseline. However, we have applied a robust method for estimation that utilises Gambia-wide case reporting but cannot adjust for the varying availability of PCR testing in the Gambia during early-mid 2020 that may result in relative under-counting in Wave 1.

We have added these results to the text as follows and also mentioned these limitations in the discussion.

“To assess whether the increased infection hazard during Omicron was due to a longer interval between infections (and therefore increased waning of immunity) an additional analysis was conducted. The time elapsed between consecutive infection episodes was calculated, with a date of first infection imputed by probability sampling for those sero-positive at baseline. Infection status was categorised as: “No prior infections”, “1 prior infection within 90 days”, “1 prior infection 90 or more days ago”, “2 or more prior infections, with the most recent within 90 days”, and “2 or more prior infections, with the most recent 90 or more days ago”. Compared to participants with no prior infections, participants with 1 prior infection in the last 90 days had a decreased infection hazard during the Delta (aHR 0.29, 95% CI 0.12 – 0.67) but not during the Omicron wave (aHR 0.73, 95% CI 0.28 – 1.91, Table S1). Similarly, for participants with 1 prior infection that occurred more than 90 days ago there was a decreased infection hazard during the Delta (aHR 0.28, 95% CI 0.19 – 0.43) but not during the Omicron wave (aHR 0.75, 95% CI 0.50 – 1.14, Table S1).

There remained strong evidence of interaction between the incidence of infection, prior infection status and the dominant variant ($p < 0.001$).”

10. In S7 and S8 the authors look at the association between potential index case factors and the risk of transmission. Would it be possible to include prior infection or vaccination status in these tables, thus getting at a reduction in transmissibility following immunity-conferring events?

Thank you for the suggestion. We have included index case serology in our SAR and HCIR models as suggested (serology acting as a marker of prior infection and vaccination status).

We find weak evidence of a lower odds of transmission from index case’s serologically positive compared to those negative in the HCIR analysis (OR 0.47, 95% CI 0.21 – 0.99, $p = 0.05$) but not in the SAR analysis (OR 0.57, 95% CI 0.24 – 1.25, $p = 0.16$, Tables S8 & S10). However, when we adjust for the serological status of the exposed household contacts this association is no longer present. We find these results interesting and suspect that given the small number of secondary transmissions detected we are underpowered to truly detect the expected protective effect of prior exposure to SARS-CoV-2.

We have included these results in the text, as well as including potential limitations in the discussion.

Discussion:

11. Remove “is” from line 239 “Africa is may be”

Thank you - we have corrected this.

Reviewer #1 (Remarks on code availability):

The code is well laid out in GitHub with a clear ReadMe. The code is to be run in numeric order, which is great and all figure code is present.

Reviewer #2 (Remarks to the Author):

Jarju and Wenlock et al conduct a prospective SARS-CoV-2 cohort study of 52 households in the Gambia over 52 weeks between March 2021 and June 2022, with weekly SARS-CoV-2 RT-PCR and 6-monthly SARS-CoV-2 serology. They have estimated high attack rates during the Delta and Omicron BA.1 waves, and symptomatic proportions in an African population with a low (about 14%) vaccine coverage. They have found prior infections are associated with reductions in infection risks during the delta predominance, and 2+ prior infections are required to develop protection against early omicron lineages. While this longitudinal study provides useful information on SARS-CoV-2 infection, symptomatic infection and protection provided by prior infections, certain questions remain to be addressed. Additionally, conducting further investigations are recommended to enhance the manuscript (details below).

1. “We aimed to recruit 50–70 households, assuming a median household size of seven.” - Justifications on the aimed sample size are required.

We have included a sentence justifying the sample size “We aimed to recruit 50–70 households, assuming a median household size of seven. This allowed for the incidence to be estimated with a 12% absolute precision (+/- 6%) and would provide an 80% power (at alpha = 0.05) to detect a 20% difference in absolute incidence risk between adults and children.”

2. “A cox proportional hazards model was fitted for the assessment of SARS-CoV-2 incidence with the Anderson-Gill extension accounting for clustering by individual (repeated measures) and household. This allowed for a varying baseline hazard over time, as observed with waves of SARS-CoV-2” – This is not clear. What does “clustering by individual (repeated measures)” refer to? Clarify for what purpose the Anderson-Gill extension is used.

Thank you – we have tried to provide clarity on this.

“The Anderson-Gill extension allowed for participants to have recurrent events with robust estimation accounting for within-person correlation as well as intra-household correlation given the household-based sampling employed here.”

The AG model allows for participants to have more than one event (e.g., be repeatedly infected) and incorporates an adjustment for the fact that a participant’s risk of infection is unique to them (i.e., there is an association between having been infected before, and risk of future infection).

3. “Calendar period was included in regression through stratification in all adjusted models (mitigating the non-proportionality) and as an interaction term with prior infection status and age” – The stratification and the interaction terms should not be in one model (the same model) because they could result in a similar effect (allow for varying risks of the exposure between periods). If I understand this sentence correctly, revisions on analyses are needed.

Thank you for pointing out this error. We had incorrectly stratified and used an interaction term for the variant-specific models. We have corrected this in the methods and updated our analyses (Table 2, Table S4). None of our conclusions have changed. The updated code is available via the GitHub page.

4. While I find the results of protection provided by prior infections and the variations by variants interesting, the analyses could be improved by conducting a deeper investigation, e.g., how the protection varies in relation to the time elapsed from the prior infection to the present infection.

Please see our response to Point 9 from Reviewer 1.

5. Table 2 footnote, “**Calculated through the incorporation with an interaction term between the variable of interest and period with adjustment for prior infection status, age, household size and vaccination status, stratified by period.” – same with comment 3.

We have removed the “stratified by period” to keep this in line with our updated modelling.

6. “444 participants underwent RT-PCR testing within one serial interval of the index case positive RT-PCR, of which 35 tested positive (a Secondary Attack Rate of 7.9%)” – What is the serial interval used in the study? Please state. Also, does the SAR vary by the variants or period?

Thank you for pointing out this omission. The serial interval in this analysis was taken to be a maximum of 14 days. We have now added this detail.

With regards to the SAR/HCIR varying by variant, this can be found in Tables S8 and Table S9 and is referenced in the text. No association between the SAR/HCIR and variant was identified.

7. Discussion: “Despite this, we estimated an attack rate of at least 57% during the 4th wave caused primarily by Omicron BA.1, with individuals requiring immunity from ≥ 2 prior infections to provide protection against infection during this first Omicron surge. The substantial antibody immune escape demonstrated by Omicron lineage viruses likely underpins this higher likelihood of breakthrough infections” – To support the above discussion, addition of analyses to investigate the reinfection risk by variants (delta vs omicron) is suggested.

Thank you for this suggestion. Unfortunately, calendar period (our proxy for infecting variant) violates the proportional hazards assumption ($p=7 \times 10^{-6}$) required to incorporate it as a fixed effect (and thus extract a hazard ratio). To incorporate this into our modelling, we include period through stratification (mitigating the non-proportionality) but this does not allow us to get further information. As a result, we are unable to make direct modelling comparisons between Delta and Omicron variants, with regards to their incidence.

We have updated our descriptive data to better describe the different re-infection risks of Delta and Omicron. In participants with a previous infection, 26.7% were re-infected during Delta (95% CI 21.6 – 32.2) whereas 56.3% were re-infected during Omicron (95% CI 50.3 – 62.2). We believe this inclusion is the best we can do to address this point given the limitations noted above.

8. Discussion: “Many studies have reported that young children are protected from severe COVID-19 and more likely to experience mild or asymptomatic infection (ref 25)” – the authors mention “many studies”, yet cite one article. While I understand that this article may be a review, I suggest adding references of the studies that provide data in support of this discussion. If such data are not found, suggest revising the sentence.

Thank you for highlighting this. We have included additional references to support our statement.

9. “Attack rates were 45% during the Delta period (7th July 2021 to 4th December 2021) and 57% during the Omicron period (after 4th December 2021).” – provide the confidence intervals.

Thank you for highlighting this oversight. We have now included these CIs.

10. “There were 270 first infections and 111 re-infections (99 second and 12 third infections) during study follow-up.” – same with comment 7, suggest adding analyses to investigate reinfection risk by variants (delta vs omicron).

Please see our response to point 7 above.

REVIEWERS' COMMENTS

Reviewer #1 (Remarks to the Author):

The authors have done a wonderful job incorporating the provided edits and suggestions. I have no additional concerns.

Reviewer #1 (Remarks on code availability):

The code remains well presented.